# Near-Field Seismic Motion: Waves, Deformations and Seismic Moment

**Bogdan Felix Apostol**

National Institute of Earth's Physics, 077125 Magurele, Romania; afelix@theory.nipne.ro

**Abstract:** The tensorial force acting in a localized seismic focus is introduced and the corresponding seismic waves are derived, as solutions of the elastic wave equation in a homogeneous and isotropic body. The deconvolution of the solution for a structured focal region is briefly discussed. The far-field waves are identified as $P$ and $S$ seismic waves. These are spherical-shell waves, with a scissor-like shape, and an amplitude decreasing with the inverse of the distance. The near-field seismic waves are spherical-shell waves, decreasing with the inverse of the squared distance. The amplitudes and the polarizations of the near-field seismic waves are given. The determination of the seismic-moment tensor and the earthquake parameters from measurements of the $P$ and $S$ seismic waves at Earth's' surface is briefly discussed. Similarly, the mainshock generated by secondary waves on Earth's surface is reviewed. The near-field static deformations of a homogeneous and isotropic half-space are discussed and a method of determining the seismic-moment tensor from epicentral near-field (quasi-) static deformations in seismogenic regions is presented.

**Keywords:** seismic tensorial force; far-field seismic waves; near-field seismic waves; seismic mainshock; quasi-static deformations

**MSC:** 35Q74; 86A15; 86A17; 86A22

## 1. Introduction

The near-field seismic ground motion is of great importance for its potentially damaging effects in epicentral regions of shallow earthquakes [1–5]. In this respect, the near-field seismic waves play the main role. At the same time, an equally important role is played by the (quasi-) static deformations produced on Earth's surface by a continuous accumulation of energy in shallow seismic foci, not necessarily resulting in an earthquake. Consequently, the near-field seismic motion is a complex subject, which requires the solution of both the elastic wave equation and elastic equilibrium (static deformations). Besides technical difficulties in getting such solutions, an important starting point is a realistic force acting in a seismc focus. Apart from the intrinsic interest in the solution itself, we we may use this solution for getting information about the focal parameters and the seismic mechanism in the focus. Such subjects are discussed in the present paper.

We start by introducing the tensorial force density acting in a seismic focus localized both in space and time (which may produce an earthquake called herein an elementary earthquake [6,7]). This is an important novelty point, because the tensorial force introduced herein is written in a covariant form, which is independent of the reference frame. In addition, it gives a vanishing total force and torque, as required by physical conditions. The deconvolution needed for a structured focus is briefly discussed. We present the solution of the Navier-Cauchy elastic wave equation with this tensorial point force in a homogeneous and isotropic body, and give information about the necessary regularization procedure employed in getting this solution [7]. The solution provides both the far-field $P$ and $S$ seismic waves and the near-field seismic waves. This is another novelty point, because the solution is obtained in compact, covariant form, without resorting to Stokes double-couple

procedure. The *P* wave is longitudinal, while the *S* wave is transverse. In the current seismological literature the *P* wave is called "primary" wave, while the *S* wave is called "secondary" wave (see, for instance [8,9]). We prefer to call them collectively "primary" waves, and use the term "secondary" waves for the mainshock. Indeed, once arrived at Earth's surface, these primary waves generate wave sources localized on the surface, which, in turn, produce secondary waves, according to Huygens principle. These secondary waves were computed, which is another novelty point [7]. They look like an abrupt wall with a long tail, propagating on Earth's surface and lagging behind the primary waves: it corresponds to the seismic mainshock recorded in seismograms. The far-field seismic waves can be used for determining the energy, the magnitude and the other earhquake parameters, as well as for determining the tensor of the seismic moment [10]). This procedure is briefly discussed here. Next, this paper is focused on the solution of the elastic equlibrium equation with the tensorial force in a homogeneous and isotropic body, and discusses the (quasi-) static elastic deformations produced on Earth's surface [6]. A special procedure of estimating the seismic moment and other focal parameters from measurements of the (quasi-) static crustal deformations made on Earth's surface is presented.

## 2. Tensorial Focal Force-Structure Factor

As it is well known, the elastic wave equation is conveniently solved with a force source localized both in space and time [8,11]. The corresponding solution is called the fundamental solution. Therefore, we assume a seismic focus placed at $R_0$, where a force source appears at the moment of time $t_0$, lasting for a short time. The corresponding force density is written as $s_{(R_0 t_0)}\delta(R - R_0)\delta(t - t_0)$, where the factor $s_{(R_0,t_0)}$ may include differential operators acting upon the variables $R$ and $t$, besides other components, arising from physical requirements (e.g., for satisfying dimensionality requirements). Let us denote the fundamental solution by $u_{(R_0 t_0)}(R - R_0, t - t_0)$ (usually called the Green function). Now, let us assume that the seismic focus has a structure, both in space and time. This structure may be represented as a linear superposition of localized sources, i.e., the force density is represented as

$$F(R,t) = \sum_i s_{(R_i t_i)}\delta(R - R_i)\delta(t - t_i) \ . \tag{1}$$

It is easy to see that the solution corresponding to the source $F(R,t)$ is given by the convolution

$$U(R,t) = \sum_i u_{(R_i t_i)}(R - R_i, t - t_i) = $$
$$= \int dR' dt' u_{(R' t')}(R - R', t - t') \sum_i \delta(R' - R_i)\delta(t' - t_i) \ . \tag{2}$$

By deconvoluting this equation, we may find out the structure of the seismic focus. The deconvolution is made by fitting the series of fundamental solutions $u_{(R_i t_i)}(R - R_i, t - t_i)$ to $U(R,t)$, where $R_i$, $t_i$ and $s_{(R_i t_i)}$ are fitting parameters.

The tensorial force density acting in a localized seismic focus is [6,7]

$$F_i(\mathbf{R},t) \quad = \quad M_{ij}T\delta(t - t_0)\partial_j\delta(\mathbf{R} - \mathbf{R}_0) \ , \tag{3}$$

where $M_{ij}$ is the (symmetric) tensor of the seismic moment, $i, j$ denote cartesian coodinates and $T$ is the (short) duration of the seismic activity in the focus. We call the earthquakes produced by this force elementary earthquakes. We note that the force density given by this equation leads to a vanishing total force and a vanishing torque. It is a representation of what is called usually the double-couple force [9] (p.60, exercise 3.6). The problem of determining the seismic waves produced by this force is similar to the Stokes problem with the force source $f_i T\delta(t - t_0)\delta(R - R_0)$ [12], where the force components $f_i$ are replaced by the operator $M_{ij}\partial_j$. Since this operator does not commute with the coordinates, we cannot

simply apply it to the Stokes solution, such that we need to rederive the solution for the force source given by Equation (3).

### 3. Seismic Waves

The elastic wave equation for a homogeneous and isotropic body (Navier-Cauchy equation) is

$$\ddot{\mathbf{u}} - c_t^2 \Delta \mathbf{u} - (c_l^2 - c_t^2) grad\, div \mathbf{u} = f \ , \tag{4}$$

where $u$ is the displacement, $c_{l,t}$ are the velocities of the longitudinal and transverse waves and $f_i = F_i/\rho$, with $F_i$ given by Equation (3) and $\rho$ the density of the body. The solution of this equation can be obtained by using the well-known Helmholtz decomposition $u = grad\Phi + curl A$, $div A = 0$ and $f = grad\phi + curl H$, $div H = 0$, where the potentials satisfy the Poisson equations $\Delta\phi = div f$, $\Delta H = -curl f$ and the wave equations $\ddot{\Phi} - c_l^2 \Delta\Phi = \phi$, $\ddot{A} - c_t^2 \Delta A = H$. These equations are solved by means of the Kirchhoff formula for retarded radiation, e.g., by using

$$\Phi(\boldsymbol{R}, t) = \frac{1}{4\pi c_l^2} \int d\mathbf{R}' \frac{\phi(t - |\mathbf{R} - \mathbf{R}'|/c_l, \mathbf{R}')}{|\mathbf{R} - \mathbf{R}'|} \ . \tag{5}$$

In applying this formula, redundant terms appear in the potentials $\Phi$ and $H$, caused by the singular derivative of the modulus function $|\mathbf{R} - \mathbf{R}'|$. This ambiguity is similar to the unphysical constant potential produced by the solution of the Poisson equation inside a sphere with a surface electrical charge. The elimination of these unphysical contributions requires a regularization of the solution [13]. The regularized solution $u = u^n + u^f$ consists of near-field displacement waves

$$u_i^n = -\frac{T}{4\pi\rho c_t^2} \frac{M_{ij}x_j}{R^3} \delta(t - R/c_t) +$$

$$+ \frac{T}{8\pi\rho R^3} \left( M_{jj}x_i + 4M_{ij}x_j - \frac{9M_{jk}x_ix_jx_k}{R^2} \right) \cdot \tag{6}$$

$$\cdot \left[ \frac{1}{c_l^2} \delta(t - R/c_l) - \frac{1}{c_t^2} \delta(t - R/c_t) \right]$$

and the far-field displacement waves

$$u_i^f = -\frac{T}{4\pi\rho c_t^3} \frac{M_{ij}x_j}{R^2} \delta'(t - R/c_t) +$$

$$- \frac{T}{4\pi\rho} \frac{M_{jk}x_ix_jx_k}{R^4} \left[ \frac{1}{c_l^3} \delta'(t - R/c_l) - \frac{1}{c_t^3} \delta'(t - R/c_t) \right] \ , \tag{7}$$

where $R$ stands for $|\boldsymbol{R} - \boldsymbol{R}_0|$ and $t$ for $t - t_0$ [7]. In these equations the $\delta(t - R/c_{l,t})$ may be viewed as a function $h(t - R/c_{l,t})$ with the support of the order $\Delta t = T$ ($\Delta R_{l,t} = c_{l,t}T$) and magnitude $1/T$, where $\Delta R_{l,t}$ are of the order of the dimension of the focus (with volume $\simeq l^3$). Similarly, the magnitude of the function $h'(t - R/c_{l,t})$ is of the order $1/T^2$.

The far-field waves given by Equation (7) are spherical-shell waves propagating with velocities $c_{l,t}$, with longitudinal and transverse polarizations, respectively, with a scissor-like shape; their amplitudes go like $1/R$ for $R \gg l$. A qualitative sketch of these waves, together with the mainshock is shown in Figure 1. These waves correspond to the $P$ (longitudinal) and $S$ (transverse) seismic waves, generated by an elementary earthquake. We call them primary waves. It is convenient to introduce the unit vector $n = \boldsymbol{R}/R$ along the propagation direction and the notations $M_{ii} = M_0$ (the trace of the tensor of the seismic moment), $M_i = M_{ij}n_j$ (the seismic-moment vector) and $M_4 = M_{ij}n_in_j$ (the unit quadratic

form of the seismic-moment tensor). The amplitudes of the far-field waves (as given by Equation (7)) can then be written as

$$\mathbf{v}_l^f = \frac{1}{4\pi\rho c_l^3 TR} M_4 \mathbf{n} \ , \ \ \mathbf{v}_t^f = \frac{1}{4\pi\rho c_t^3 TR}(\boldsymbol{M} - M_4\mathbf{n}) \ . \tag{8}$$

Similarly, the near-field waves look like spherical shells propagating with velocities $c_{l,t}$, with mixed polarizations. Their amplitudes of the near-field waves (Equation (6)) can be written as

$$\boldsymbol{v}^n(c_l) = \frac{1}{8\pi\rho c_l^2 TR^2}[(M_0 - 9M_4)\boldsymbol{n} + 4\boldsymbol{M}] \ ,$$

$$\boldsymbol{v}^n(c_t) = -\frac{1}{8\pi\rho c_t^2 TR^2}[(M_0 - 9M_4)\boldsymbol{n} + 6\boldsymbol{M}] \tag{9}$$

for waves which propagate with velocities $c_{l,t}$. The longitudinal and transverse parts of these waves are

$$v_l^n(c_l) = \frac{1}{8\pi\rho c_l^2 TR^2}(M_0 - 5M_4)\boldsymbol{n} \ , \ \ v_t^n(c_l) = \frac{1}{2\pi\rho c_l^2 TR^2}(\boldsymbol{M} - M_4\boldsymbol{n}) \tag{10}$$

and

$$v_l^n(c_t) = -\frac{1}{8\pi\rho c_t^2 TR^2}(M_0 - 3M_4)\boldsymbol{n} \ , \ \ v_t^n(c_t) = -\frac{3}{4\pi\rho c_t^2 TR^2}(\boldsymbol{M} - M_4\boldsymbol{n}) \ . \tag{11}$$

These amplitudes decrease like $1/R^2$ for $R \gg l$.

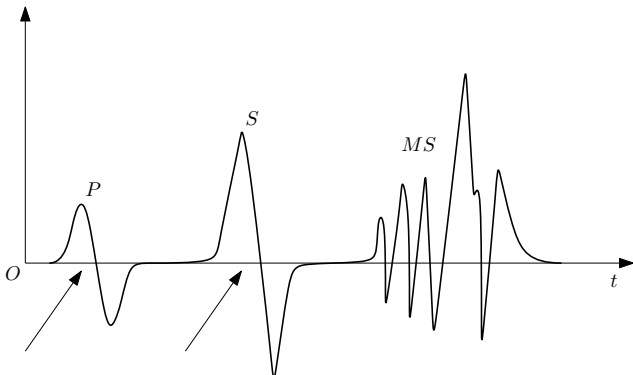

**Figure 1.** A qualitative sketch of scissor-like $P$ and $S$ seismic waves (indicated by arrows) and the seismic main shock ($MS$), vs. time $t$.

A spherical-shell wave has a thickness of the order $\Delta R = cT$, where $c$ is a generic notation for wave velocities. It affects a circular epicentral region with radius $d$ on Earth's surface. For a focus placed at depth $h$ the radius $d$ is given by $(h + \Delta R)^2 = h^2 + d^2$, i.e., $d \simeq \sqrt{2h\Delta R}$ (for $\Delta R \ll h$). For instance, for $h = 100$ km and $\Delta R = 3$ km we get $d \simeq 24$ km. This epicentral displacement lasts approximately $\Delta t \simeq \Delta R/c$, e.g., $\Delta t \simeq 1$ s for $c = 3$ km/s. Thereafter, the spherical-shell wave (primary wave) propagates on Earth's surface with a circular wavefront. The points on Earth's surface where the seismic wave arrives become sources of secondary elastic waves, propagating back in the Earth and on Earth's surface. Their cummulative effect on Earth's surface look like an abrupt wall with a long tail [7]. Specifically, the surface displacement in cylindrical coordinates behaves like $u_{r,\varphi} \sim r/(c^2\tau^2 - r^2)^{3/2}$, $u_z \sim 1/r(c^2\tau^2 - r^2)^{3/2}$, where $r$ is the radial coordinate on Earth's surface (assumed a plane surface) and $\tau$ is the time from the moment when the wave touched the epicentre. This is the mainshock, as recorded in typical seismograms. A primary wave propagates on Earth's surface with a (non-uniform) velocity larger than the elastic-wave velocity of the mainshock, such that there exists a time delay between the arrival of the primary wave and the arrival of the mainshock, which laggs behind the

primary wave. The formulae given above for the amplitudes of these secondary waves are valid for a limited range of epicentral distances centered on $r \sim h$. Their singularities at $c\tau = r$ are smoothed out by the non-uniform velocity, e.g., $c^2\tau^2 - r^2 \longrightarrow h^2$ for $c\tau - r \simeq 0$ and the time delay is of the order $h^2/cr$ for $h \ll r$ [7].

## 4. Seismic Moment

The amplitudes of the primary $P$ and $S$ waves (Equation (8)) measured at Earth's surface can be used to determine the tensor of the seismic moment and earthquake parameters like energy, magnitude, fault orientation, the magnitude of the fault slip, and to estimate the duration of the seismic activity in the focus and the dimension of the focus [10]. This is achieved by using the energy conservation in the propagation of the seismic waves, the work done by the focal forces and the Kostrov representation of a shearing fault. The results are comparable with the results produced by the currently used methods [14–19]. For instance, by using this method, the estimated magnitude of the Vrancea earthquake of 28 October 2018 was 5.3, while the current method gave 5.5 (as reported by the Institute of Earth's Physics, Magurele in Romanian Earthquake Catalogue, ROMPLUS (2018)) [20,21]. In addition, this information can be used to get an estimate of the near-field waves, according to Equation (9) (the Kostrov representation leads to a vanishing trace $M_0 = 0$ of the seismic-moment tensor). Similarly, the method can be applied to explosions, where the tensor of the seismic moment is diagonal ($M_{ij} = -M\delta_{ij}$) [10]. For orientative purposes it is worth giving here a recipe for a qualitative estimate of these parameters. The duration of the seismic activity in the focus can be estimated by $T = \sqrt{2Rv}/c$, where $v$ is a generic amplitude of the primary waves measured at distance $R$ form the focus on Earth's surface, and $c$ is a generic elastic-wave velocity (e.g., $c = 3 - 7$ km/s). The volume of the focal region is $V = \pi(2Rv)^{3/2}$, the released energy is of the order $E = \mu V$, where $\mu$ is the Lame coefficient, and the magnitude of the seismic moment is $(M_{ij})^{1/2} = 2\sqrt{2}E$. The well-known Hanks-Kanamori relationship $\lg(M_{ij})^{1/2} = 1.5M_w + 16.1$ provides the magnitude $M_w$ [10].

Another method of getting information about the seismic-moment tensor is given here, by using the quasi-static deformations produced by a tensorial focal force in near-field epicentral zones of the seismogenic regions.

A continuous accumulation of tectonic stress may be gradually discharged, to some extent and with intermittence, causing quasi-static crustal deformations of Earth's surface in seismogenic zones [22–28]. Measurements of these deformations may give, besides qualitative information about the seismic activity, an estimation of the depth of the focus and the focal volume, as well as an opportunity of estimating the tensor of the seismic moment for a shearing fault.

The static deformations produced by a tensorial point force density $\mathbf{f}$ in a homogeneous isotropic elastic half-space are given by the equation of elastic equilibrium

$$\Delta\mathbf{u} + \frac{1}{1 - 2\sigma}\operatorname{grad}\operatorname{div}\mathbf{u} = -\frac{2(1 + \sigma)}{E}\mathbf{f} \ , \tag{12}$$

where $\mathbf{u}$ is the displacement vector (with components $u_i$, $i = 1, 2, 3$), $E$ is the Young modulus and $\sigma$ is the Poisson ratio. The components of the force density are given by

$$f_i = M_{ij}\partial_j\delta(\mathbf{r} - \mathbf{r}_0) \ , \tag{13}$$

where $\mathbf{r}_0$ is the position of the focus and $M_{ij}$ is the tensor of the seismic moment. It is convenient to write $\bar{\mathbf{f}} = -[2(1 + \sigma)/E]\mathbf{f}$ and $\overline{M}_{ij} = -[2(1 + \sigma)/E]M_{ij}$ (reduced force and reduced seismic moment). Equation (12) is solved for a half-space $z < 0$, with free surface $z = 0$, the position of the focus being $\mathbf{r}_0 = (0, 0, z_0)$, $z_0 < 0$ (epicentral frame); we use the radial coordinate $\rho = (x^2 + y^2)^{1/2}$ with in-plane coordinates $x$, $y$ and notations $x_1 = x$,

$x_2 = y$, $x_3 = z$. The components of the displacement vector of the surface $z = 0$ are given by [6]

$$2\pi \cdot u_\alpha = -\overline{M}_{\alpha\beta} I_\beta^{(1)} + \overline{M}_{\alpha 3} I^{(0)} -$$

$$-\tfrac{1}{2}\overline{M}_{\beta\gamma}\partial_\beta\partial_\gamma[2\sigma I_\alpha^{(3)} - z_0 I_\alpha^{(2)}] - \tag{14}$$

$$-z_0\overline{M}_{3\beta}\partial_\beta I_\alpha^{(1)} + \tfrac{1}{2}\overline{M}_{33}[2\sigma I_\alpha^{(1)} + z_0 I_\alpha^{(0)}] \ ,$$

and

$$2\pi \cdot u_3 = -\tfrac{1}{2}\overline{M}_{\alpha\beta}\partial_\beta[(1 - 2\sigma)I_\alpha^{(2)} + z_0 I_\alpha^{(1)}] +$$

$$+z_0\overline{M}_{3\alpha} I_\alpha^{(0)} + \tfrac{1}{2}\overline{M}_{33}[(1 - 2\sigma)I^{(0)} - z_0\tfrac{\partial}{\partial z_0}I^{(0)}] \ , \tag{15}$$

where

$$I^{(0)} = -\tfrac{z_0}{r^3} \ , \quad I^{(1)} = \tfrac{1}{r} \ , \quad I_\alpha^{(2)} = -\tfrac{x_\alpha}{r(r+|z_0|)} \ , \tag{16}$$

$$I_\alpha^{(3)} = -\tfrac{x_\alpha}{r+|z_0|} \ ,$$

$I_\alpha^{(n)} = \partial_\alpha I^{(n)}$ ($n = 0, 1, 2, 3$) and $r = (\rho^2 + z_0^2)^{1/2}$; we use $\alpha$, $\beta$, $\gamma = 1, 2$. The solution can be compared with previous results [29], obtained by using particular cases of the Mindlin solution.

The components $u_\alpha$ given by Equation (14) are vanishing for $\rho \longrightarrow 0$ and go like $1/\rho^2$ for $\rho \longrightarrow \infty$; they have a maximum value for $\rho$ of the order $|z_0|$. The component $u_3$ goes like $1/z_0^2$ for $\rho \longrightarrow 0$ and $1/\rho^2$ for $\rho \longrightarrow \infty$. It is convenient to give these displacement components for $\rho$ close to zero, i.e., in the seismogenic zone (close to a presumable epicentre). We get

$$u_\alpha = \tfrac{1}{16\pi}\left[4(1 - 2\sigma)\overline{M}_{33} - (3 + 2\sigma)\overline{M}_0\right]\tfrac{x_\alpha}{|z_0|^3} +$$

$$+\tfrac{1}{8\pi}(1 - 2\sigma)\tfrac{\overline{M}_{\alpha\beta}x_\beta}{|z_0|^3} + \dots \ ,$$

$$u_3 = \tfrac{1}{8\pi z_0^2}\left[2(3 - 2\sigma)\overline{M}_{33} - (1 + 2\sigma)\overline{M}_0\right] + \tag{17}$$

$$+\tfrac{\overline{M}_{3\alpha}x_\alpha}{2\pi|z_0|^3} + \dots \ ,$$

where $\overline{M}_0 = \overline{M}_{ii}$ is the trace of the tensor $\overline{M}_{ij}$.

A simplified numerical estimation of the unknowns (components of the seismic moment) can be obtained as follows. We assume $M_0 = 0$ (as for a shearing fault), replace all the components of the seismic-moment tensor in Equation (17) by a mean value $\overline{M}$ and average over the orientation of the vector $\rho$; we denote the resulting $u_3$ by $u_v$ (vertical component) and introduce $u_h$ (horizontal component) by $u_h = (u_1^2 + u_2^2)^{1/2}$; we get approximately

$$u_h \simeq \frac{(1 - 2\sigma)|\overline{M}|}{4\pi}\frac{\rho}{|z_0|^3} \ , \quad u_v \simeq \frac{(3 - 2\sigma)\overline{M}}{4\pi z_0^2} \ ; \tag{18}$$

hence, we get immediately the depth of the focus

$$|z_0| \simeq \frac{1 - 2\sigma}{3 - 2\sigma}|u_v|/(\partial u_h/\partial\rho) \tag{19}$$

and the mean value $\overline{M} = 4\pi z_0^2 u_v/(3 - 2\sigma)$ of the (reduced) seismic moment. Making use of $\overline{M}_{ij} = -[2(1 + \sigma)/E]M_{ij}$, we have

$$M_{av} \simeq -\frac{2\pi E}{(1 + \sigma)(3 - 2\sigma)}z_0^2 u_v \tag{20}$$

for the mean value $M_{av}$ of the seismic moment $M_{ij}$ (Equation (18)). Since the small displacement values $u_h$, $u_v$ may be affected by errors, a mean value of the seismic moment may be viewed as satisfactory. For $M_{av} = 10^{22}$ dyn $\cdot$ cm (which would correspond to an earthquake with magnitude $M_w = 4$ by the Hanks-Kanamori law lg $M_{av} = 1.5 M_w + 16.1$), Young modulus $E = 10^{11}$ dyn/cm$^2$, $\sigma = 0.25$ and depth $\mid z_0 \mid = 100$ km we get a vertical displacement $u_v \simeq 1$ μm; we can see that the static surface displacement is, indeed, very small. A reliable determination of such quasi-static diplacemenents may raise difficulties. The seismicity accounts for a very small fraction of crustal deformation [30].

A rough estimate for the elastic energy stored by the static deformation is given by $\mathcal{E} \simeq 4\pi z_0^2 E \mid u_v \mid \simeq 2(1 + \sigma)(3 - 2\sigma)|M_{av}|$; it is also given by $\mathcal{E} \simeq \mu V$, where $\mu$ is the Lame coefficient and $V$ is the focal volume ($\mu = E/2(1 + \sigma)$; the other Lame coefficient is $\lambda = E\sigma/(1 - 2\sigma)(1 + \sigma)$); making use of the approximations introduced above, we get $V \simeq 8\pi(1 + \sigma)z_0^2 \mid u_v \mid$. For $\mid z_0 \mid = 100$ km and $u_v = 1$ μm ($\sigma = 0.25$) we get a volume $V \simeq 10^5 \pi$ m, i.e., a linear dimension $l \simeq 500$ m (noteworthy, a static deformation may diffuse, such that the corresponding focal volume is larger than the focal volume of a sudden earthquake discharge). Similarly, from Equation (17) we get an estimate $u_{ij} \sim V / \mid z_0 \mid^3$ for the surface strain; using the numerical data above, it is extremely small, of the order $10^{-10}$.

For more specific information we make use of the general results of static deformations [6]; the displacement components given by Equation (17) can be written in a general form (for $M_0 = 0$) as

$$
u_i = \{[2(3 - 2\sigma)\overline{M}_4^{(n)} - (9 - 10\sigma)\overline{M}_4^{(nv)}]n_i -
$$
$$
-4\overline{M}_4^{(n)}v_i + (1 - 2\sigma)\overline{M}_{ij}v_j\}\frac{1}{8\pi z_0^2} \; , \tag{21}
$$

where

$$
\mathbf{n} = (x_\alpha, z - z_0) / \mid z_0 \mid , \; \mathbf{v} = (x_\alpha, z) / \mid z_0 \mid ,
$$
$$
\overline{M}_4^{(n)} = \overline{M}_{ij}n_i n_j , \; \overline{M}_4^{(nv)} = \overline{M}_{ij}n_i v_j \; ; \tag{22}
$$

in Equations (21) and (22) we retain only contributions linear in $x_\alpha$ and in the limit $z \to 0$. Within these restrictions the form given by Equation (21) is unique. In these equations

$$
\overline{M}_i = \overline{M}_{ij}v_j \simeq \frac{\overline{M}_{i\alpha}x_\alpha}{\mid z_0 \mid} \tag{23}
$$

are the components of a vector and

$$
\overline{M}_4^{(n)} \simeq 2\overline{M}_3 + \overline{M}_{33} , \; \overline{M}_4^{(nv)} \simeq \overline{M}_3 \tag{24}
$$

are scalars. Taking the scalar product $\mathbf{nu} \simeq u_3$ in Equation (21), we get

$$
\overline{M}_4^{(n)} = \frac{4\pi z_0^2 u_3 + 4(1 - \sigma)\overline{M}_3}{3 - 2\sigma} \; ; \tag{25}
$$

inserting this $\overline{M}_4^{(n)}$ and $\overline{M}_4^{(nv)} \simeq \overline{M}_3$ in Equation (21) we are led to

$$
u_\alpha = \frac{1 - 2\sigma}{3 - 2\sigma}\frac{x_\alpha}{\mid z_0 \mid}u_3 + \frac{1 - 2\sigma}{8\pi z_0^2}\overline{M}_\alpha \tag{26}
$$

(and the identity $u_3 = u_3$). This equation gives

$$
\overline{M}_\alpha = 8\pi z_0^2 \left( \frac{1}{1 - 2\sigma}u_\alpha - \frac{1}{3 - 2\sigma}\frac{x_\alpha}{\mid z_0 \mid}u_3 \right) \tag{27}
$$

(and $M_\alpha = -[E/2(1+\sigma)]\overline{M}_\alpha$) as functions of the measured quantities $u_\alpha$, $u_3$ and $x_\alpha$; $\overline{M}_4^{(nv)}$ and $\overline{M}_4^{(n)}$ are given by Equations (24) and (25) as functions of $u_3$ and the parameter $\overline{M}_3$, which remains undetermined. This is the maximal information provided by measuring the static displacement in a seismogenic zone; the parameter $z_0$ can be estimated from Equation (19).

Further on, we assume that the components $M_\alpha$ of the vector **M** are determined from data, according to Equation (27); the component $M_3$ will be determined shortly. The scalars $M_4^{(nv)} \simeq M_3$ and $M_4^{(n)}$ are given by Equations (24) and (25), respectively; they depend on the parameter $M_3$. Parameters $z_0$ (focus depth) and the focal volume $V$ remain undetermined. Order-of-magnitude estimations given above (Equation (19) and below) may be used to this end.

In order to determine the components of the seismic moment we use the Kostrov expression derived for a shearing fault [10]; it is given by

$$M_{ij} = M^0(s_i a_j + s_j a_i) \ , \ i, j = 1, 2, 3 \ , \tag{28}$$

where $M^0 = 2\mu V$ and $s_i$, $a_i$ are the components of two orthogonal unit vectors **s** and **a**: **s** is normal to the fault plane and **a** is directed along the fault displacement (fault sliding). We can see that Equation (22) implies $M_0 = M_{ii} = 0$. We assume that the measured data of the static displacement satisfy this condition. In addition, we assume that $M^0$ is a known parameter.

We introduce the scalar products $A = \mathbf{av}$ and $B = \mathbf{sv}$ and write

$$A\mathbf{s} + B\mathbf{a} = \mathbf{m} \ , \ B\mathbf{s} + A\mathbf{a} = \mathbf{v} \tag{29}$$

from Equation (28), where $\mathbf{m} = \mathbf{M}/M^0$; we solve this system of equations for **s** and **a** with the conditions $s^2 = a^2 = 1$, $\mathbf{sa} = 0$. We note that Equation (28) is invariant under the symmetry operations $\mathbf{s} \longleftrightarrow \mathbf{a}$ and $\mathbf{s}, \mathbf{a} \longleftrightarrow -\mathbf{s}, -\mathbf{a}$ (and $\mathbf{s} \longleftrightarrow -\mathbf{a}$); consequently, it is sufficient to retain one solution of the system of Equation (29) (it has multiple solutions), all the others being given by these symmetry operations. We get

$$\mathbf{s} = \tfrac{A}{A^2-B^2}\mathbf{m} - \tfrac{B}{A^2-B^2}\mathbf{v} \ , \ \mathbf{a} = -\tfrac{B}{A^2-B^2}\mathbf{m} + \tfrac{A}{A^2-B^2}\mathbf{v} \tag{30}$$

and

$$A^2 + B^2 = m^2 = v^2 \ , \ 2AB = v^2 m_4 \ , \tag{31}$$

where $m_4 = \mathbf{mv}/v^2 = M_{ij}v_i v_j/v^2 M^0$. From $m^2 = v^2$ we get the component $M_3$, as given by

$$M_3^2 = \left(M^0\right)^2 v^2 - M_\alpha^2 \ ; \tag{32}$$

we may take

$$A = v\sqrt{\frac{1 + \sqrt{1 - m_4^2}}{2}} \ , \ B = sgn(m_4) \cdot v\sqrt{\frac{1 - \sqrt{1 - m_4^2}}{2}} \tag{33}$$

as a solution of the system of Equation (31); this solves the problem of determining the seismic moment from the measurements of the surface static displacement. From Equation (28) the seismic-moment tensor is given by

$$M_{ij} = \frac{M^0}{v^2(1 - m_4^2)}\left[m_i v_j + m_j v_i - m_4\left(m_i m_j + v_i v_j\right)\right] \ ; \tag{34}$$

the vector **v** is known from Equation (22) ($z \to 0$, $v = \rho/ \mid z_0 \mid$) and the vector **m** is known from Equations (27) and (32) (with $z_0$ and $M^0$ as known parameters); the scalar $m_4$ is given by $m_4 = M_\alpha v_\alpha/v^2 M^0$. The component $M_3$ does not enter the expression of $m_4$; it is

included in $M_{ij}$. The quadratic form $M_{ij}x_ix_j = const$ is a hyperbola; its asymptotes indicate the fault plane (vector **s**) and the fault slip (vector **a**).

The isotropic case $M_{ij} = -M^{is}\delta_{ij}$, where $M^{is} = 2(2\mu + \lambda)V$, implies a surface displacement

$$\mathbf{u} = \frac{M^{is}(1+\sigma)}{4\pi z_0^2 E}[(3 - 10\sigma)\mathbf{n} - (3 - \sigma)\mathbf{v}] \ , \tag{35}$$

the vector **M** being given by $\mathbf{M} = -M^{is}\mathbf{v}$. The energy can be estimated as $\mathcal{E} = M^{is}/2 = 4\pi z_0^2 E \mid u_v \mid$, which leads to a focal volume $V = [4\pi(1 + \sigma)(1 - 2\sigma)/(1 - \sigma)]z_0^2 \mid u_v \mid$.

## 5. Concluding Remarks

The solutions of the elastic wave equation and the equation of elastic equilibrium in a homogeneous and isotropic body are presented, for a tensorial point force acting in a seismic focus localized both in space and time. The solutions exhibit both the far-field elastic waves, identified as primary *P* and *S* seismic waves, and the static deformations, discussed herein mainly on Earth's surface. The mainshock of secondary waves produced by the primary waves arrived at Earth's surface is briefly discussed, as well as the procedure of determining the tensor of the seismic moment and the other earthquake parameters (like energy, magnitude, fault orientation, fault slip, duration of the seismic activity in focus and an estimated dimension of the seismic focus). A procedure of extracting such information from the measurements of the crustal (quasi-) static deformations made on Earth's surface is also presented. We may envisage that such a procedure, in spite of its challenging difficulties, may become applicable.

The seismic waves and the static deformations discussed in this paper are derived for a homogeneous and isotropic elastic body. While this may be viewed as a reasonable assumption for a large-scale description, it is a serious limitation for the near-field scale, where the local inhomogeneities (local site effects) play an important part.

**Funding:** This work was funded by the Romanian Ministry of Research and Innovation, Program Nucleu 2919, Research Grant #PN19-08-01-02/2019.

**Institutional Review Board Statement:** Not applicable.

**Informed Consent Statement:** Not applicable.

**Data Availability Statement:** Not applicable.

**Acknowledgments:** The author is indebted to the colleagues in the Institute of Earth's Physics, Magurele-Bucharest, for many enlightening discussions. This work was partially carried out within the Program Nucleu 2019, funded by Romanian Ministry of Research and Innovation, Research Grant #PN19-08-01-02/2019. Data used for the Vrancea region have been extracted from the Romanian Earthquake Catalog, ROMPLUS, 2018.

**Conflicts of Interest:** The author declares no conflict of interest

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
