# Peer review of "Near-Field Seismic Motion: Waves, Deformations and Seismic Moment"

_axioms, doi:10.3390/axioms11080409_

Round 1

Reviewer 1 Report

The reviewer is sorry that no figures or tables are included in this manuscript.

In other words, it is unfortunate that the reviewer judges this manuscript to be rejected because it is not well-formed as a scientfic and technological paper.

Reviewer 2 Report

he article needs some corrections. The author writes about P and S seismic waves in lines 40 and 41, indicating that they are primary waves. In seismology, P waves , as the first group of waves recorded on the seismogram, are called primary waves, while P waves , as appearing on the seismogram as the second group of waves, are called secondary waves.

So it is not clear at this point what the author meant by the content contained in line 40.

Perhaps already at this point it would be appropriate to introduce an explanation for P-waves (longitudinal) and S-waves (transverse). This is the nomenclature he uses later in the article. 

There is no explanation of all the parameters used in the formulas. I recommend that the author analyze the article and supplement the text with explanations of parameters. There are many such places in the article.

It is not understood what the author means by the phrase that he presents a note.  Is it a note or an article?

Reviewer 3 Report

This could be a potentially useful contribution to this journal, although not exactly about axiomatic science. 

The Author should however better describe the novelty of his approach in 1-2 sentences in the abstract and in a longer way in the last paragraph of the Introduction referring there to the existing, classical literature.

For the benefit of the readers it is suggested to add 1 - 3 simple illustrations showing a statement of the problem and perhaps some other elements (e.g. "scissor-like" waves etc.)

Correct English language in some places See for example line 123: "They cummulative effect (...)" etc.

Round 2

Reviewer 1 Report

The reviewer confirmed that the authors had not responded to any of the reviewers' initial comments.

The authors explain that they added figures, but no figures have been added.

It is hard to say that the response to the second comment was also supported.

Therefore, the reviewer rejects this revised manuscript.